# Zero-Dose, Under-Immunized, and Dropout Children in Nigeria: The Trend and Its Contributing Factors over Time

**DOI:** 10.3390/vaccines11010181

**Published:** 2023-01-14

**Authors:** Ryoko Sato

**Affiliations:** Harvard T.H. Chan School of Public Health, 90 Smith Str., Boston, MA 02120, USA; rsato@hsph.harvard.edu

**Keywords:** parental knowledge, childhood immunization, Pakistan, vaccine hesitancy, infectious disease

## Abstract

Introduction: This study analyzes the trend of prevalence of, and factors contributing to, children with incomplete vaccination status, namely zero-dose, under-immunized, and dropout children, over time from 2003 to 2018 in Nigeria, one of the countries with the highest number of children with incomplete vaccination. Methods: Nigeria Demographic and Health Survey data from 2003 to 2018 were analyzed to calculate the prevalence of children with incomplete vaccination status by geographical zone over time and to investigate the factors contributing to the change in the prevalence of such children over time based on the Blinder–Oaxaca decomposition analysis. Results: The prevalence of children with incomplete vaccination status substantially decreased from 2003 to 2018 in most of zones in Nigeria. Rural areas and the northern zones had consistently higher prevalence of children with incomplete vaccination status than urban areas and the southern zones. It was identified that mothers’ education and the household wealth level explained the reduction in the prevalence of zero-dose and under-immunized children, but the degree of contribution of each factor varied by zone and place of residence, i.e., urban or rural. Both the mother’s education and household wealth level only weakly contributed to the reduction in the number of dropout children. Discussions and conclusion: Future studies should explore further how to improve the vaccination coverage in Nigeria over time. Examples of topics for future study include other contributing factors beyond education and wealth level, differential factors influencing the reduction in the number of children with incomplete vaccination status by area of residence (urban vs. rural), why the reduction in the number of dropout children is not explained by either education or wealth, and the significant contributors to the reduction in the number of such children.

## 1. Introduction

Vaccination saves lives, and it does so effectively and cost-effectively [1,2].

However, 25 million children were still under-immunized globally, according to UNICEF, in 2022, and over 17 million of children did not receive any basic vaccines across the world, according to WHO/UNICEF, in 2020 [3,4]. Furthermore, the distribution of such under-immunized children is highly unequal. Over sixty percent of such under-immunized children reside in 10 countries, one of which is Nigeria [3]. 

Nigeria has been struggling to provide children with vaccination. For example, the vaccination coverage against the third dose of the diphtheria tetanus toxoid and pertussis vaccine (DTP3) was merely 50%, and almost one in five children (19%) never received any vaccination in 2018 [5]. Because Nigeria is the most populous country in sub-Saharan Africa, this high rate of children with incomplete vaccination status translates to the large number of children with such status. Indeed, Nigeria is one of the five countries that account for two-thirds of all the children who did not receive any vaccines in the world, according to Save the Children, in 2022 [6].

To advance the universal immunization coverage, it is crucial to understand the barriers to immunization. Barriers to immunization can manifest in various forms, and the types of barriers might be different. In particular, this paper focuses on three distinguished forms of incomplete immunization: zero-dose, under-immunized, and dropout. This paper studies the progress in the reduction in the number of children with incomplete vaccination status, including zero-dose, under-immunized, and dropout children, and examines the factors contributing to reductions in the number of children with such incomplete vaccination status over time in Nigeria.

## 2. Methods

### 2.1. Data

The Nigeria Demographic and Health Survey (NDHS), conducted in 2003, 2008, and 2018, was used for the analysis [7]. The NDHS is a national household survey with representative samples at the national and state levels. The NDHS datasets contain various information on health-seeking behaviors, including the vaccination status among children aged between 0 and 36 months old within the 5 years prior to the survey, as well as sociodemographic characteristics.

### 2.2. Outcome Variables

There are three main outcome variables in this study: (1) whether a child is zero-dose, (2) whether a child is under-immunized, and (3) whether a child is a dropout. This paper defines these three statuses as incomplete vaccination status. It follows the conventional definition of each variable from the Global Alliance for Vaccines and Immunization, the Vaccine Alliance (GAVI). A child was defined as a zero-dose child if she/he did not receive the first dose of the DPT or pentavalent vaccine (DPT1/Penta1). A child was under-immunized if she/he did not receive the third dose of the DPT or pentavalent vaccine (DPT3/Penta3). Additionally, a child was a dropout if she/he received DPT1/Penta1 but not DPT3/Penta3. We used the vaccination records of children from the NDHS data to construct these variables. 

### 2.3. Covariates

The paper used various sociodemographic characteristics to evaluate the correlations between the main outcome variables. These included the mother’s age, child’s age, mother’s education level, household wealth level, and the place of residence (rural/urban), taken from the NDHS as covariates for the correlational analysis. 

#### Analysis

Firstly, the prevalence of zero-dose children, under-immunized children, and dropout children over time was calculated across Nigeria in urban and rural areas, respectively, and across geographical zones. Nigeria is divided into 6 zones: North-Central, North-East, North-West, South-East, South-South, and South-West. 

Then, the sociodemographic factors associated with children being zero-dose, under-immunized, and dropout were evaluated using ordinary least-squares (OLS) regression. Based on this correlational analysis, the trend of important sociodemographic factors was also examined over time in each geographic zone. 

Finally, Blinder–Oaxaca decomposition analysis was conducted to explore the factors contributing to the change in the prevalence of children with incomplete vaccination status (zero-dose, under-immunized, and dropout) over time. Details of the conceptual framework of Blinder–Oaxaca decomposition can be found in the paper by Jann (2008) [8].

### 2.4. Results

Figure 1 presents the prevalence of zero-dose, under-immunized, and dropout children over time in Nigeria. The prevalence of children with these incomplete vaccination statuses consistently decreased over time from 2003 to 2018: the percentage of zero-dose children reduced from 61% to 40%, the percentage of under-immunized children reduced from 79% to 57%, and the percentage of dropout children reduced from 46% to 28%. While this trend is consistent regardless of the place of residence, whether urban or rural areas, the prevalence of such children with incomplete vaccination status was much higher in rural areas than in urban areas at any given time. For example, the prevalence of zero-dose children in 2003 was 42% in urban areas and 69% in urban areas; the prevalence of under-immunized children was 64% in urban areas and 86% in rural areas; and the prevalence of dropout children was 38% in urban areas and 53% in rural areas.

Figure 2 presents the prevalence of zero-dose children by zone in urban and rural areas, respectively. Overall, the prevalence of children with incomplete vaccination status decreased from 2003 to 2018 in most zones in both urban and rural areas. The only exception was the prevalence of zero-dose children in the rural South-West zone, which increased constantly from 27% in 2003 to 37% in 2018.

Table 1 presents the determinants of the rates of children with incomplete vaccination status using the overall sample and by urban and rural area, respectively. Generally, the older age of mothers, older age of children, and higher education level of mothers were all significantly and negatively correlated with the prevalence of incomplete vaccination status (zero-dose, under-immunized, and dropout) in the total sample, as well as in urban and in rural area, respectively. Similarly, a higher level of wealth level of household was negatively correlated with the prevalence of incomplete vaccination status. However, the correlation was stronger in rural areas than in urban areas. The exception was that there was little to no correlation between the wealth level and children being dropouts in urban areas.

Figure 3 presents the change in the proportions of mothers with no educational attainment and of households in the poorest quintile in urban and rural areas, respectively, in each zone. In rural areas, the proportions of mothers with no education and of households in the poorest quintile both decreased consistently over time from 2003 to 2008 in all zones except the North-Central zone. Unlike in rural areas, on the other hand, the trend of the proportions of mothers with no educational attainment and of households in the poorest quintile were not uniform in urban areas. For example, the proportion of mothers with no education increased in the South-South zone in urban areas, while the other zones observed its reduction. The proportion of the households in the poorest quintile decreased in North-East and North-West zones, but other zones (North-Central, South-East, South-South, and South-West) observed an increase.

Table 2 presents the result of the Blinder–Oaxaca decomposition for each incomplete vaccination status by area of residence (urban/rural). In both urban and rural areas, the changes in the proportions of zero-dose, under-immunized, and dropout children over time were all largely due to unexplained factors. Among the explained factors, the reductions in the prevalence of zero-dose children and under-immunized children were largely due to the improvement in education in both urban and rural areas. The improvement in the household wealth level was a significant explanatory factor for the zero-dose and under-immunized children in rural areas but not in urban areas. The reduction in the prevalence of dropout children was largely explained by the improvement in the educational level of mothers in urban areas, but this was not the case in rural areas. On the other hand, the reduction in the prevalence of dropout children was significantly explained by the improvement in the household wealth level in rural areas.

Figure 4 presents the contribution of the improvement in mothers’ education and household wealth, respectively, with the overall reduction in the prevalence of each incomplete vaccination status. Overall, a similar pattern of the contribution of each factor was observed for the reductions in the number of zero-dose children and under-immunized children. In urban areas, the improvement in the education level of mothers contributed positively to the reduction in the numbers of zero-dose children and under-immunized children in most zones except North-West and South-West zones. The improvement of the wealth level of households also contributed to the reduction in the numbers of both zero-dose and under-immunized children in most zones, but the magnitude of the contribution tended to be smaller than that of education in urban areas. In rural areas, the improvements in both mothers’ education and household wealth contributed to the reduction in the numbers of zero-dose and under-immunized children in most or all of the zones more consistently than in urban areas. However, the magnitude of the contributions of both education and wealth appeared to be smaller in rural areas than in urban areas.

On the other hand, little or none of the improvements in either education or wealth contributed to the reduction in the number of dropout children in both urban and rural areas. In urban areas, only the North-Central zone showed that education and wealth explained the reduction in the number of dropout children. In rural areas, more zones highlighted the education and wealth as significant contributors to the reduction in the number of dropouts, but the magnitude of the contribution was much smaller than the contribution to the reduction in the numbers of zero-dose and under-immunized children.

## 3. Discussion

This paper estimated the prevalence of children with incomplete vaccinated status over time in Nigeria, namely zero-dose, under-immunized, and dropout children, and evaluated the factors that have contributed to the reduction in the prevalence of such children, focusing mainly on the improvements in education and wealth levels. The analyses were based on three waves of the Nigeria Demographic and Health Survey (NDHS), with nationally representative samples, between 2003 and 2018.

Overall, it was found that the prevalence of children with incomplete vaccination status, including zero-dose, under-immunized, and dropout, was much higher in rural areas than in urban areas in any survey year. In the 15 years between the initial survey year in 2003 and the last one in 2018, the prevalence of children with any of the incomplete vaccination statuses decreased substantially in both urban and rural areas, but the gap in such prevalence did not narrow between urban and rural areas over time.

The pattern of the reduction in the prevalence of children with incomplete vaccination status varied by zone. The southern zones consistently had a lower prevalence of children with incomplete vaccination status than the northern zones. While most of the zones observed a reduction of the prevalence of children with incomplete vaccination status, the increase in the prevalence of zero-dose children was observed in the South-West zone.

The likelihood of children being either zero-dose or under-immunized was correlated with the lower educational attainment of mothers and lower household wealth level, but the association was stronger in rural areas than in urban areas. While the likelihood of children being dropouts was strongly associated with the lower educational attainment of mothers, it was only weakly explained by the wealth level in both urban and rural areas.

Because education and wealth were both significant explanatory variables for the lower likelihood of children having incomplete vaccination status, it was evaluated whether the trend in the proportion of children with incomplete vaccination status coincided with the trends in mothers’ educational attainment and the wealth level across zones. While both the proportion of women with no education and the proportion of households in the poorest wealth quintile decreased in most of zones over time, this trend did not appear to be associated with the reduction in the number of children with incomplete vaccination status across zones. The higher reduction in the number of children with incomplete vaccination status in a zone was not correlated with a greater reduction in the number of mothers with no educational attainment or that of households in the poorest wealth quintile in that zone.

Based on the Blinder–Oaxaca decomposition analysis, this paper found that the substantial reduction in the number of children with incomplete vaccination status was primarily due to unexplained factors. Among the explained factors, the improvement in mothers’ education level was the dominant contributor to the reduction in the prevalence of children with incomplete vaccination status, except for dropout children in rural areas. The improvement in the household wealth level was also a significant contributor to the reduction in the prevalence of children with incomplete vaccination status in rural areas but not in urban areas.

In general, the improvement in mothers’ education level contributed more to the reduction in the numbers of zero-dose and under-immunized children than the improvement in the household wealth level in urban areas. In rural areas, on the other hand, the magnitude of the contributions of both education and wealth was smaller than it was in urban areas. The reduction in the number of dropout children was mostly explained by neither the improvement in education nor the improvement in wealth in both urban and rural areas.

Some findings of this paper, particularly the determinants of incomplete vaccination status, are consistent with findings from the literature [9]. Mahachi et al. (2022) conducted a meta-analysis on the risk factors for zero- and missed-dose children in Nigeria and found that some demographic characteristics, including the mother’s education and wealth level of the community, were risk factors. This paper advanced this analysis further to identify factors contributing to the reduction in the prevalence of such children with incomplete vaccination status, while many papers examined the factors contributing to the prevalence of incomplete vaccination but not the reduction in such factors, including Mahachi et al. (2022) [9]. Future studies should investigate other factors contributing to under-immunization, in addition to mothers’ education and wealth, including the availability of vaccines, perceptions about vaccine efficacy, such as the importance of vaccination, and access to clinics offering vaccination, as well as factors that explain the reduction in the number of children with incomplete vaccination status, especially dropout children, beyond education and the wealth level.

## 4. Conclusions

While Nigeria has observed a substantial reduction in the prevalence of children with incomplete vaccination status, the pattern of, and explanatory factors for, the reduction differ. Furthermore, little is known about the factors explaining such a reduction. This paper was one of the first steps in efforts to unveil how a country could achieve higher vaccination coverage by identifying the factors contributing to the reduction in the number of children with incomplete vaccination status. However, more detailed studies should be conducted to further explore this area of research, including other contributing factors beyond education and wealth level, differential factors influencing the reduction in the number of children with incomplete vaccination status by area of residence (urban vs. rural), why the reduction in the number of dropout children is not explained by either education or wealth, and the significant contributors to such a reduction.

## Figures and Tables

**Figure 1 vaccines-11-00181-f001:**
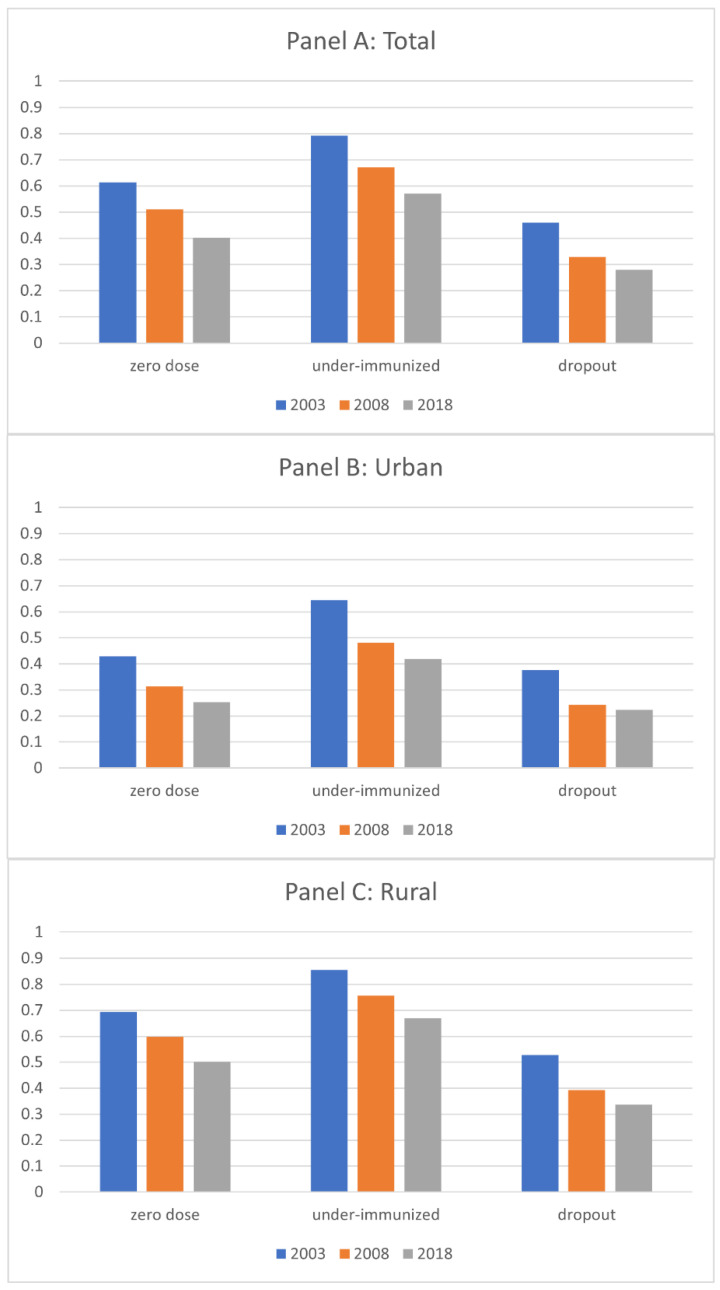
Prevalence of Children with Incomplete Vaccination Status: Zero-dose, Under-Immunized, and Dropout.

**Figure 2 vaccines-11-00181-f002:**
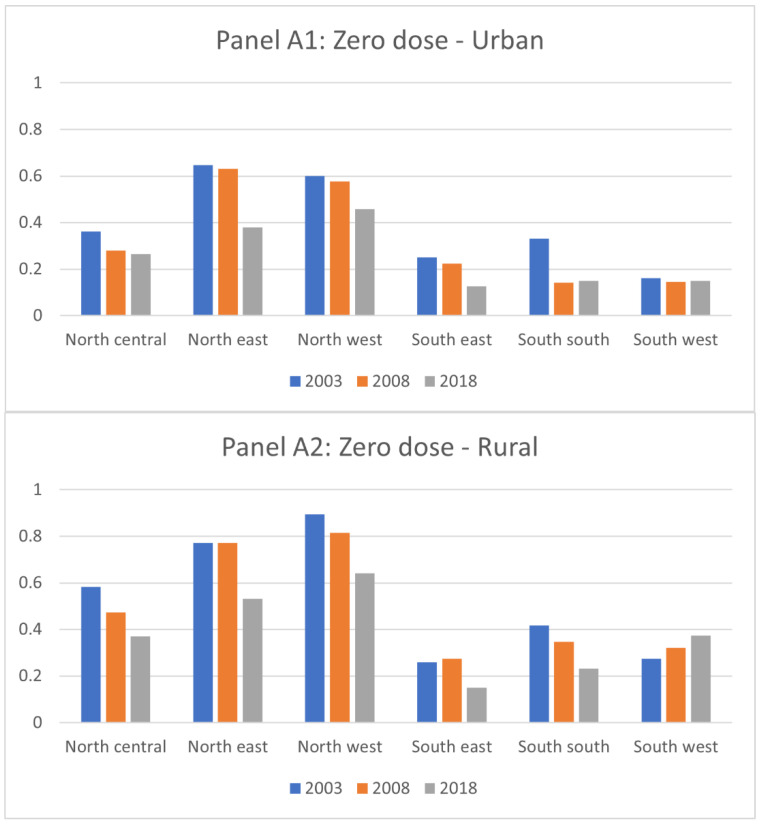
Prevalence of Children with Incomplete Vaccination Status: Zero-dose, Under-Immunized, and Dropout by Zone.

**Figure 3 vaccines-11-00181-f003:**
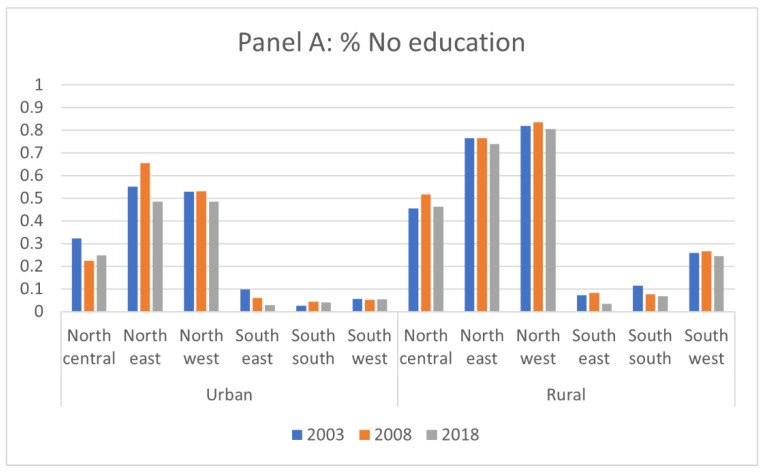
Trend of proportions of mothers with no educational attainment and of households in the poorest quintile over time by zone.

**Figure 4 vaccines-11-00181-f004:**
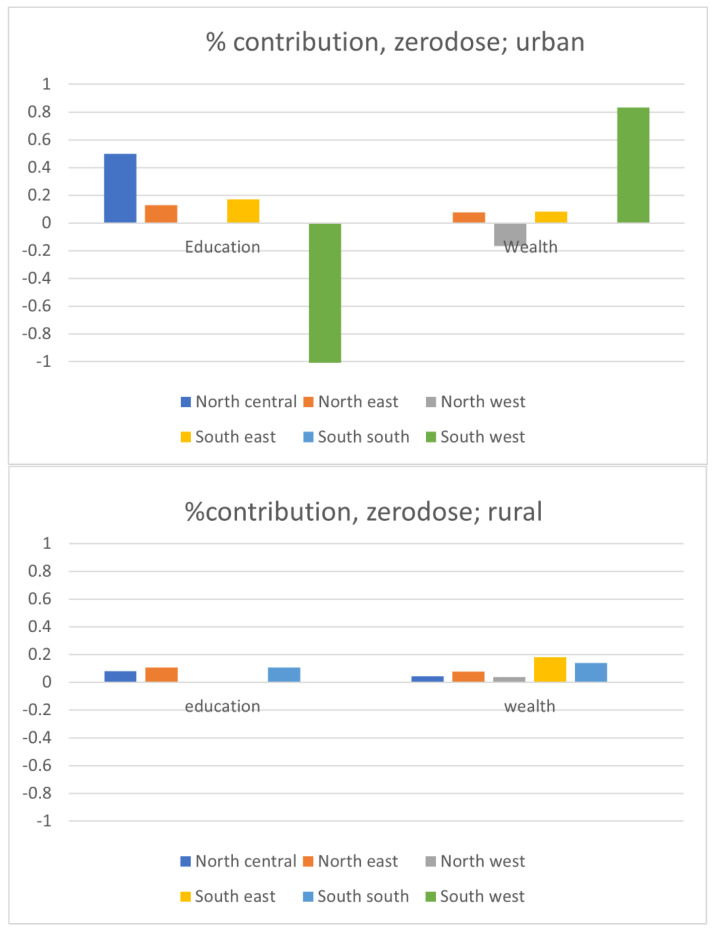
Contributions of the improvements in education and wealth to the reduction in the number of children with incomplete vaccination status by zone.

**Table 1 vaccines-11-00181-t001:** Correlation between the likelihood of children having incomplete vaccination status and sociodemographic factors.

	Zero-Dose			Under-Immunized			Dropout		
	Total	Urban	Rural	Total	Urban	Rural	Total	Urban	Rural
	(1)	(2)	(3)	(4)	(5)	(6)	(7)	(8)	(9)
Mother’s age	−0.005 **	−0.004 ***	−0.005 ***	−0.004 ***	−0.005 ***	−0.004 ***	−0.004 ***	−0.003 ***	−0.004 ***
	(0.000)	(0.001)	(0.000)	(0.000)	(0.001)	(0.000)	(0.000)	(0.001)	(0.001)
Child’s age	−0.026 ***	−0.036 ***	−0.022 ***	−0.041 ***	−0.061 ***	−0.032 ***	−0.052 ***	−0.054 ***	−0.051 ***
	(0.002)	(0.003)	(0.002)	(0.002)	(0.003)	(0.002)	(0.002)	(0.004)	(0.003)
Mother’s education (comparison = no education)									
Primary education	−0.294 ***	−0.291 ***	−0.293 ***	−0.217 ***	−0.259 ***	−0.203 ***	−0.127 ***	−0.152 ***	−0.120 ***
	(0.005)	(0.011)	(0.006)	(0.005)	(0.011)	(0.006)	(0.009)	(0.016)	(0.010)
Secondary education	−0.444 ***	−0.426 ***	−0.457 ***	−0.381 ***	−0.418 ***	−0.367 ***	−0.209 ***	−0.228 ***	−0.203 ***
	(0.005)	(0.009)	(0.007)	(0.005)	(0.010)	(0.006)	(0.008)	(0.014)	(0.010)
Tertiary education	−0.493 ***	−0.469 ***	−0.537 ***	−0.479 ***	−0.496 ***	−0.507 ***	−0.256 ***	−0.271 ***	−0.272 ***
	(0.009)	(0.013)	(0.016)	(0.009)	(0.014)	(0.014)	(0.012)	(0.016)	(0.019)
Household wealth level (comparison = poorest)									
Poorer	0.005	−0.030	0.013	−0.011	−0.025	−0.005	−0.027 **	−0.008	−0.022
	(0.008)	(0.025)	(0.008)	(0.008)	(0.027)	(0.007)	(0.014)	(0.035)	(0.015)
Medium	0.032 ***	−0.002	0.040 ***	0.019 ***	0.039	0.021 ***	0.015	0.082 **	0.013
	(0.007)	(0.024)	(0.008)	(0.007)	(0.026)	(0.007)	(0.013)	(0.035)	(0.014)
Richer	−0.048 ***	−0.049 ***	−0.043 ***	−0.043 ***	−0.015	−0.043 ***	−0.045 ***	0.022	−0.050 ***
	(0.006)	(0.019)	(0.007)	(0.006)	(0.020)	(0.006)	(0.010)	(0.026)	(0.011)
Richest	−0.111 ***	−0.113 ***	−0.104 ***	−0.100 ***	−0.067 ***	−0.103 ***	−0.065 ***	0.007	−0.074 ***
	(0.007)	(0.018)	(0.009)	(0.007)	(0.020)	(0.008)	(0.010)	(0.025)	(0.013)
Rural (comparison = urban)	0.048 ***	0.000	0.000	0.061 ***	0.000	0.000	0.049 ***	0.000	0.000
	(0.005)	(.)	(.)	(0.005)	(.)	(.)	(0.007)	(.)	(.)
Survey year (comparison = 2003)									
2008	−0.073 ***	−0.074 ***	−0.073 ***	−0.092 ***	−0.120 ***	−0.080 ***	−0.128 ***	−0.120 ***	−0.135 ***
	(0.007)	(0.012)	(0.008)	(0.006)	(0.013)	(0.007)	(0.011)	(0.016)	(0.015)
2018	−0.174 ***	−0.139 ***	−0.192 ***	−0.195 ***	−0.203 ***	−0.193 ***	−0.227 ***	−0.185 ***	−0.258 ***
	(0.007)	(0.012)	(0.008)	(0.007)	(0.013)	(0.008)	(0.011)	(0.016)	(0.015)
_cons	0.993 ***	0.972 ***	1.048 ***	1.145 ***	1.202 ***	1.177 ***	0.835 ***	0.743 ***	0.917 ***
	(0.012)	(0.027)	(0.013)	(0.012)	(0.029)	(0.012)	(0.021)	(0.037)	(0.025)
N	47,439	14,390	33,049	47,439	14,390	33,049	23,925	9620	14,305
r2	0.259	0.201	0.218	0.240	0.191	0.194	0.090	0.071	0.076

Notes: ** Significant at 5%, *** Significant at 1%.

**Table 2 vaccines-11-00181-t002:** Blinder–Oaxaca Decomposition Analysis of the reduction in the number of the children with incomplete vaccination status.

	Zero-Dose			Under-Immunized			Dropout		
	Total	Urban	Rural	Total	Urban	Rural	Total	Urban	Rural
	(1)	(2)	(3)	(4)	(5)	(6)	(7)	(8)	(9)
Differential									
Prediction_1 (year = 2003)	0.592 ***	0.438 ***	0.681 ***	0.779 ***	0.654 ***	0.851 ***	0.457 ***	0.385 ***	0.532 ***
	(0.007)	(0.011)	(0.008)	(0.006)	(0.011)	(0.006)	(0.011)	(0.015)	(0.016)
Prediction_2 (year = 2018)	0.413 ***	0.277 ***	0.479 ***	0.585 ***	0.444 ***	0.654 ***	0.293 ***	0.231 ***	0.335 ***
	(0.004)	(0.006)	(0.005)	(0.004)	(0.007)	(0.004)	(0.005)	(0.007)	(0.006)
Difference	0.179 ***	0.161 ***	0.202 ***	0.193 ***	0.210 ***	0.197 ***	0.164 ***	0.154 ***	0.197 ***
	(0.008)	(0.013)	(0.009)	(0.007)	(0.013)	(0.008)	(0.012)	(0.016)	(0.017)
Explained									
Education	0.024 ***	0.051 ***	0.016 ***	0.025 ***	0.054 ***	0.016 ***	−0.000	0.016 ***	−0.005
	(0.003)	(0.005)	(0.004)	(0.003)	(0.005)	(0.003)	(0.002)	(0.003)	(0.003)
Wealth	0.005 ***	0.002	0.009 ***	0.004 ***	0.001	0.010 ***	0.000	0.000	0.006 **
	(0.001)	(0.002)	(0.002)	(0.001)	(0.001)	(0.002)	(0.001)	(0.001)	(0.002)
Rural	−0.002 ***	0.000	0.000	−0.003 ***	0.000	0.000	−0.006 ***	0.000	0.000
	(0.000)	(.)	(.)	(0.001)	(.)	(.)	(0.001)	(.)	(.)
Total	0.027 ***	0.053 ***	0.025 ***	0.027 ***	0.055 ***	0.026 ***	−0.006 *	0.016 ***	0.001
	(0.004)	(0.005)	(0.004)	(0.003)	(0.005)	(0.004)	(0.003)	(0.004)	(0.004)
Unexplained									
Education	0.000	−0.002	0.001	−0.026 ***	−0.008	−0.058 ***	−0.002	−0.006	−0.017
	(0.007)	(0.007)	(0.015)	(0.008)	(0.008)	(0.016)	(0.008)	(0.009)	(0.017)
Wealth	−0.011 ***	−0.038 **	−0.005	−0.005 *	−0.030 **	−0.002	−0.012	−0.025	−0.006
	(0.003)	(0.016)	(0.003)	(0.003)	(0.014)	(0.003)	(0.010)	(0.028)	(0.008)
Rural	0.026 **	0.000	0.000	0.017	0.000	0.000	0.026 *	0.000	0.000
	(0.011)	(.)	(.)	(0.010)	(.)	(.)	(0.014)	(.)	(.)
_cons	0.136 ***	0.148 ***	0.180 ***	0.181 ***	0.192 ***	0.231 ***	0.158 ***	0.169 ***	0.219 ***
	(0.016)	(0.021)	(0.018)	(0.016)	(0.021)	(0.018)	(0.024)	(0.033)	(0.025)
Total	0.151 ***	0.108 ***	0.176 ***	0.167 ***	0.155 ***	0.171 ***	0.170 ***	0.137 ***	0.196 ***
	(0.007)	(0.012)	(0.009)	(0.007)	(0.013)	(0.007)	(0.012)	(0.016)	(0.017)
N	22,399	7513	14,886	22,399	7513	14,886	12,229	5129	7100

Notes: * Significant at 10%, ** Significant at 5%, *** Significant at 1%.

## Data Availability

The data used in this study is publicly available and can be found at https://www.dhsprogram.com/Countries/Country-Main.cfm?ctry_id=30&c=Nigeria&Country=Nigeria&cn=&r=1, accessed on 12 October 2022.

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
