# Peer review of "Zero-Dose, Under-Immunized, and Dropout Children in Nigeria: The Trend and Its Contributing Factors over Time"

_vaccines, 2023, doi:10.3390/vaccines11010181_

Round 1

Reviewer 1 Report

MS: Zero-dose, under-immunized, and dropout children in Nigeria: Trend and its contributing factors over time

Major revisions

In this work, the author analyzes the trend of prevalence and contribution factors of children with incomplete vaccination status: zero-dose, under-immunized, and dropout over time from 2003 to 2018 in Nigeria.

-        My main concern is a possible lack of “originality” of this paper, due to this is almost similar to the work of Mahachi et al., 2022

-        Despite author declares “Some findings of this paper, in particular, determinants of incomplete vaccination status, are consistent with findings from the literature (Mahachi et al., 2022). However, this paper is one of the first attempt to identify contributing factors to the reduction of prevalence of such children with incomplete vaccination status. Future study should investigate more on factors that explain the reduction of children with incomplete vaccination status, especially dropout children, beyond education and wealth level”… Author must remark carefully what new/original findings are supported in the present paper with respect to the work reported by Mahachi et al., 2022.

-        Introduction and discussion must be improved.

-        Figures, tables and presentation also must be improved, in order to the reader can follow result and data more affordable.

-        English diction and grammar need extensive editing.

Author Response

MS: Zero-dose, under-immunized, and dropout children in Nigeria: Trend and its contributing factors over time

Major revisions

In this work, the author analyzes the trend of prevalence and contribution factors of children with incomplete vaccination status: zero-dose, under-immunized, and dropout over time from 2003 to 2018 in Nigeria.

-        My main concern is a possible lack of “originality” of this paper, due to this is almost similar to the work of Mahachi et al., 2022

-        Despite author declares “Some findings of this paper, in particular, determinants of incomplete vaccination status, are consistent with findings from the literature (Mahachi et al., 2022). However, this paper is one of the first attempt to identify contributing factors to the reduction of prevalence of such children with incomplete vaccination status. Future study should investigate more on factors that explain the reduction of children with incomplete vaccination status, especially dropout children, beyond education and wealth level”… Author must remark carefully what new/original findings are supported in the present paper with respect to the work reported by Mahachi et al., 2022.

[rs] Thank you for pointing this out, and my apologies for not making it clear in the manuscript. I now revised the section as follows:

“Some findings of this paper, in particular determinants of incomplete vaccination status, are consistent with findings from the literature (Mahachi et al., 2022). Mahachi et al. (2022) conducted a meta-analysis on risk factors for zero- and missed-dose children in Nigeria and found that some demographic characteristics including mother’s education and wealth level of community were risk factors. I advanced the analysis further in this paper to identify contributing factors to the reduction of prevalence of such children with incomplete vaccination status, while many papers examined the contributing factors to the prevalence of incomplete vaccination, not to the reduction of such factors, including Mahachi et al. (2022).”

-        Introduction and discussion must be improved.

[rs] I have revised the introduction (last paragraph) as follows:

“To advance on the universal immunization coverage, it is crucial to understand barriers to immunization. Barriers to immunization can manifest in various forms, and the type of barriers might be different. In particular, I focus on three distinguished forms of incomplete immunization: zero-dose, under-immunized, and dropout. This paper studies the progress of the reduction of children with incomplete vaccination status: zero-dose, under-immunized, and dropout, and examines contributing factors to such reductions over time in Nigeria.”

I revised the discussion (last paragraph) described below:

“Some findings of this paper, in particular determinants of incomplete vaccination status, are consistent with findings from the literature (Mahachi et al., 2022). Mahachi et al. (2022) conducted a meta-analysis on risk factors for zero- and missed-dose children in Nigeria and found that some demographic characteristics including mother’s education and wealth level of community were risk factors. I advanced the analysis further in this paper to identify contributing factors to the reduction of prevalence of such children with incomplete vaccination status, while many papers examined the contributing factors to the prevalence of incomplete vaccination, not to the reduction of such factors, including Mahachi et al. (2022). Future study should investigate more on other factors contributing to the under-immunization in addition to mother’s education and wealth, including availability of vaccines, perception about vaccine efficacy such as importance of vaccination, and access to clinic offering vaccination, as well as factors that explain the reduction of children with incomplete vaccination status, especially dropout children, beyond education and wealth level.”

-        Figures, tables and presentation also must be improved, in order to the reader can follow result and data more affordable.

[rs] Thank you. I have revised all the figures and tables for the improved presentation.

-        English diction and grammar need extensive editing.

[rs] I have asked the proofreader to improve the English throughout the paper.

Reviewer 2 Report

Major issues are the figures and tables, which are the major part of the author's paper.  The figures need to be higher resolution, split into a, b, c..., and include legends.  The tables have to be readable. I suggest adding cell margins, changing the font size, highlighting the significant results, fitting within the paper margins, and adding legends.   

Author Response

Thank you very much for the useful comments. 

For the comment on the figures and tables ("Major issues are the figures and tables, which are the major part of the author's paper.  The figures need to be higher resolution, split into a, b, c..., and include legends.  The tables have to be readable. I suggest adding cell margins, changing the font size, highlighting the significant results, fitting within the paper margins, and adding legends. ")

-  I have now revised all the figures and tables as per your advice.

Please see the revised manuscript.

Also, please see the attachment for the revisions I made, responding to the inquiry in the pdf file.

Reviewer 3 Report

- The study is well- designed and the manuscript is well- written.

- Further investigations should be carried out to highlight  the other causes that may contribute to the problem of under-immunized children rather than mother education or wealth level. Other factors like the availability of vaccines, increasing the awareness of mothers about the importance of vaccination and the availability of  hospitals or vaccination centers, should be discussed 

Author Response

Thank you for your comment. I have responded to your comment at [rs].

Further investigations should be carried out to highlight  the other causes that may contribute to the problem of under-immunized children rather than mother education or wealth level. Other factors like the availability of vaccines, increasing the awareness of mothers about the importance of vaccination and the availability of  hospitals or vaccination centers, should be discussed 

[rs] Thank you for your advice. I have now added the following at the end of discussion as per your advice;

“Future study should investigate more on other factors contributing to the under-immunization in addition to mother’s education and wealth, including availability of vaccines, perception about vaccine efficacy such as importance of vaccination, and access to clinic offering vaccination, as well as factors that explain the reduction of children with incomplete vaccination status, especially dropout children, beyond education and wealth level.”

Round 2

Reviewer 2 Report

Accept in present form

Author Response

Thank you very much for reviewing and approving the revision.